

# Runoff simulation by SWAT model using high-resolution gridded precipitation in the upper Heihe River Basin, Northeastern Tibetan Plateau

Hongwei Ruan[1,2], Songbing Zou[1,3], Zhentao Cong[4], Yuhan Wang[4], Zhenliang Yin[1], Zhixiang Lu[1], Fang Li[1,2], Baorong Xu[3]

[1]Key Laboratory of Ecohydrology of Inland River Basin, Northwest Institute of Eco-Environment and Resources, Chinese Academy of Sciences, Lanzhou, 730000, China

[2]University of Chinese Academy of Sciences, Beijing, 100049, China

[3]College of Earth Environmental Sciences, Lanzhou University, Lanzhou, 730000, China

[4]Department of Hydraulic Engineering, Tsinghua University, Beijing, 100084, China

*Correspondence to*: Songbing Zou (zousongbing@lzb.ac.cn)

**Abstract.** Precipitation stations are usually scarce and unevenly distributed in inland river basins, which restrict the application of the distributed hydrological model and spatial analysis of water balance component characteristics. This study regards the upper Heihe River Basin as a case, and daily gridded precipitation data with 3 km resolutions based on the spatial interpolation of gauged stations and the regional climate model is used to construct the soil and water assessment tool (SWAT). This study aims to validate the superiority of high-resolution gridded precipitation for hydrological simulation in data scarce regions. A scale transformation method is proposed by building virtual stations and calculating the lapse rate to overcome the defects of the SWAT model using traditional precipitation station data. The gridded precipitation is upscale from the grid to the sub-basin scale and results in accurate representation of sub-basin precipitation input data. A satisfactory runoff simulation is achieved, and the spatial variability of the water balance components is analysed. Results show that the precipitation lapse rate ranges from 40 mm/km to 235 mm/km and decreases from the southeastern to the northwestern areas; its changes trend is consistent with precipitation. The SWAT model achieves monthly runoff simulation compared with gauged runoff from 2000 to 2014; the determination coefficients are higher than 0.71, the Nash–Sutcliffe efficiencies are higher than 0.76 and the percent bias are controlled within ±15%. The meadow and sparse vegetation are the major water yield landscapes, and the elevation band at 3,500 m to 4,500 m is the major water yield area in this basin. Precipitation and evapotranspiration presented a slightly increasing trend, whereas water yield and soil water content presented a slightly decreasing trend. This finding indicates that the high-resolution gridded precipitation data well depicts its spatial heterogeneity, and scale transformation significantly promotes the application of the distributed hydrological model in inland river basins. The spatial variability of water balance components can be quantified to provide references for the integrated assessment and management of basin water resources in data scarce regions.

## 1 Introduction

With the changes in the global climate and the frequent harmful human activities, water resource shortage occurs, which restricts the midstream social development and leads to the downstream eco-environment degradation in inland river basins of Northwest China (Yang et al., 2015). Runoff is mainly generated from alpine cold mountainous regions, which significantly affects the midstream and the downstream (Gao et al., 2015). Hydrological models are widely used for the integrated assessment and management of basin water resources. Precipitation is an important input for accurate hydrological simulation, and its numerical accuracy and detailed spatial distribution are necessary (Morrissey et al., 1995; Yu et al., 2011). However, precipitation gauge stations are scarce and unevenly distributed in the alpine cold mountainous regions of Northwest China because of economy, terrain, transport and technology limitations (Lu et al., 2015). Thus, these gauge stations are difficultly represent the spatial heterogeneity of regional precipitation, leading to high uncertainty in hydrological simulation and analysis. Alternatively, this data scarce situation can be addressed using high-resolution gridded





precipitation.

Gridded precipitation based on gauge stations has been widely investigated and used to establish hydrological models. Version 6 of the global precipitation product developed by the Global Precipitation Climatology Centre has monthly resolutions of 0.5° from 1901 to 2010 (Becker et al., 2013). Li et al. (2014) and Huang et al. (2014) used the spline interpolation and trend surface methods to determine gridded precipitation in China. Yang et al. (2014) evaluated different gridded precipitations to establish a hydrological model of the Three Gorges Reservoir. Fuka et al. (2013) used the NCEP-CFSR data to construct a hydrological model for validating the accuracy of gridded precipitation. Previous studies also used the sparse meteorological stations to construct gridded precipitation in China; these stations tend to be poor representations of the amount and spatial distribution of precipitation (Shen et al., 2015). By contrast, hydrological stations can provide gauged precipitation to complete precipitation data. The regional climate model (RCM) can also supply information on spatial distribution to correct gridded precipitation. Thus, the gridded precipitation data based on the spatial interpolation of abundant gauged stations and RCM simulation can well depicts its spatial heterogeneity, which is suitable for driving a hydrological model.

The hydrological model has been increasingly used to analyse the hydrological process in Heihe River Basin (HRB), where the water resource shortage problems are typical in inland river basins of Northwest China. Soil and water assessment tool (SWAT) is a physical, semi-distributed hydrological model, which is widely used to assess and manage water resources in the upper HRB. The performance of this model relies on precipitation input parameters, namely, accuracy and spatial distribution (Tobin and Bennett., 2009). The SWAT model applied to the upper HRB is focused on model modification and hydrological process responses to climate and land use change; thus far, few studies have optimised precipitation input parameters using gridded data (Li et al., 2009, 2010, 2011; Zang et al., 2013; Yin et al., 2014). The gridded precipitation data with daily resolutions of 3 km have been developed for the HRB through the spatial interpolation of the meteorological station, the hydrological station and the RCM simulation (Yang et al., 2015; Gao et al., 2015). This high-resolution gridded precipitation is preferred for hydrological simulation and analysis. However, the SWAT model only employs precipitation data from one station closest to the centroid of each sub-basin that can be corrected by elevation band and lapse rate; as such, the current method of representing precipitation in the SWAT model is simplistic. This leads to inaccurate representation of the sub-basin precipitation input data (Masih et al., 2011). Most studies directly input gridded data into the model by building virtual stations for all grids that cannot maximise the superiority of high-resolution gridded data because most grids are ignored (Zou et al., 2016; Sood et al., 2013). Thus, a reasonable scale transformation from the grid to the sub-basin must be developed to maximise the superiority of the high-resolution gridded precipitation data in horizontal and vertical distributions.

The spatial variability of water balance components can be quantified to assess and manage water resources in the upper HRB (Yin et al., 2016). Water balance components are difficult to gauge but can be calculated by the SWAT model. From the inputs of high-resolution gridded precipitation, the SWAT model can simulate water balance components with high accuracy and detailed spatial distribution. Currently, water balance components are estimated at different scales, namely, global, regional, watershed and ecosystem levels (Thompson et al., 2011; Gregory et al., 2013; Herrmann et al., 2015; Jian et al., 2015). Thus, the spatial distribution, change trend and internal relationship of water balance components across different scales can further strengthen our understanding of the hydrological processes.

This study mainly aims to (1) assess the quality of gridded precipitation data in the upper HRB; (2) conduct scale transformation by building virtual precipitation stations to transfer gridded data into a sub-basin average and calculating precipitation lapses rates on the sub-basin scale, optimising the input parameters of precipitation; (3) assess the performance of the SWAT model by comparing the monthly runoff simulation with observed data; and (4) analysing the spatial variability and change trend of water balance components on the sub-basin, landscape and elevation band scales according to the simulation results. The subsequent sections present the general situation of the study area and the available data. Methods were developed for data processing, data evaluation, scale transformation and calculation of precipitation lapse rate. Finally, the gridded precipitation data and the hydrological simulation results were evaluated; and the spatial variability and change trend of water balance components were analysed.





## 2 Study area and data availability

### 2.1 Study area

The HRB, the second largest inland river basin in Northwest China, originates from the Qilian Mountains in the Northeastern Tibetan Plateau and flows through the middle of the Hexi Corridor, which was an important district of the ancient Silk Road (Liu et al., 2012). The upper HRB generates approximately 70% of the river flows of the entire basin, which supports the social development of the midstream and maintains the eco-environment balance of the downstream (Deng et al., 2015). This study focuses on the upper HRB, whose drainage area is approximately 10,018 km² and mostly covered by mountainous terrain (Fig. 1). The elevation of the study area ranges from 1,667 m to 5,008 m, with a mean elevation of 3,737 m. The basin outlet is monitored by the Yingluoxia hydrological station. Hydrological stations in Qilian and Zhamashike are situated in the east and west tributaries of the upper HRB, respectively (Lu et al., 2015). The flow of these two tributaries joins the mainstream at the Huangzangsi and enters the basin outlet. The basin consists of three subregions, namely, east tributary, west tributary and mainstream. The study area, which is a typical inland region with large spatiotemporal variability, experiences a dry and cold climate in winter and a moist and hot climate in summer. The mean annual precipitation varies from 200 mm to 700 mm, which tends to decrease from the southeast to the northwest and increases along with elevation; approximately 60% of the precipitation occurs in the summer. The mean annual temperature in this region ranges from −5 °C to 4 °C (Zang et al., 2013; Wu et al., 2015). The landscape follows a distinct vertical zonation and comprises the desert, steppe, shrub, coniferous forest, meadow, sparse vegetation, snow and glaciers, which vary from low to high elevations. The major soil types in the basin include felty, chestnut and alpine frost soils (Zhang et al., 2016).

### 2.2 Data availability

Data used in this study are categorised into: data on geospatial information and climate forcing for SWAT model setup; and data on gridded precipitation assessment for SWAT model validation. The meteorological data include daily temperature, sunshine hour, wind speed and relative humidity, which serve as climate forcing data. These data were downloaded form the National Meteorological Stations of China Meteorological Administration (CMA). Gridded precipitation data with daily resolutions of 3 km were used as precipitation forcing data downloaded from the Heihe Plan Science Data Centre (HPSD). Digital elevation model with 90 m resolution was downloaded from the Shuttle Radar Topography Mission (SRTM) database (Jarvis et al., 2008). The soil map of the upper HRB was derived from the China Second National Soil Survey. The vegetation map, with a measuring scale of 1:100,000, was obtained from the HPSD; the vegetation pattern boundary was adjusted relative to that of the previous version (Fig. 2). The gauged precipitation data, including 4 meteorological and 13 hydrological stations, were used to evaluate gridded precipitation (Fig. 1), which were obtained from CMA and HPSD, respectively. The daily gauged runoff data were obtained from the Hydrology and Water Resources Bureau of Gansu Province and used to validate the SWAT model.

## 3 Methods

### 3.1 SWAT model

The SWAT model is a physical, semi-distributed hydrological model that can operate under different climate conditions and land use change scenarios. The model is widely used to simulate long-term yields in large watersheds for the assessment and management of water resources (Arnold et al., 2012). The runoff transport process considered in the SWAT model includes watershed land areas that transport water to the channels and through the stream network to the watershed outlet (Neitsch et al., 2011). The basin is divided into hydrologic response units (HRUs) that integrated unique land use, soil type





and slope, which are the basic elements of hydrological calculation. The HRUs of water balance components, such as precipitation, evapotranspiration, water yield, surface flow, lateral flow and groundwater flow, were calculated (Neitsch et al., 2005). The hydrological processes simulated by the SWAT model are based on the following water balance equation:

$$SW_t = SW_0 + \sum_{i=1}^{t}(P_{day} - Q_{surf} - E_a - W_{seep} - Q_{gw}) \, , \tag{1}$$

where $SW_t$ is the final soil water content (mm), $SW_0$ is the initial soil water content on day $i$ (mm), $t$ is time (days), $P_{day}$ is the amount of precipitation on day $i$ (mm), $Q_{surf}$ is the amount of surface runoff on day $i$ (mm), $E_a$ is the amount of evapotranspiration on day $i$ (mm), $W_{seep}$ is the amount of water that enters the vadose zone from the soil profile on day $i$ (mm) and $Q_{gw}$ is the amount of return flow on day $i$ (mm).

**3.2 Gridded precipitation data**

The gridded precipitation data were spatially interpolated by using the method developed by Shen and Xiong (2015). First, a gridded analysis of daily precipitation climatology was built based on the Shepard interpolation method and mean daily gauged precipitation data from 1960 to 2014. The gauged precipitation, including 16 meteorological stations and 25 hydrological stations of the HRB, were smoothed by Fourier transformation to remove high-frequency noise precipitation caused by insufficient sampling, real extreme events and random measurement errors. Second, the regional integrated
environmental model system RCM was calibrated using the HRB observed data, which provide the spatial distribution of the precipitation lapse rate. The RCM simulation is used to correct the precipitation lapse rate of daily precipitation climatology. Third, the optimal interpolation was employed to create the gridded ratio field that is the ratio of the daily gauged precipitation to daily precipitation climatology. Finally, the gridded precipitation is calculated by multiplying the daily gridded precipitation climatology with the gridded ratio field. The gridded precipitation data with daily resolutions of 3 km
for a time series were generated over the HRB (Yang, et al., 2015; Gao et al., 2015; Qin et al., 2016).

**3.3 Virtual precipitation station**

The precipitation data are inputted to SWAT model in the form of station data, and then the precipitation station data are discretized to sub-basin scale that can be corrected by the elevation band and lapse rate (Arnold et al., 2012). Thus, the grids of gridded precipitation data are treated as virtual precipitation station, and virtual station is a commonly method that grid data input to SWAT model. (Zou et al., 2016; Sood et al., 2013; Paxian et al., 2015; Price et al., 2013). However, the SWAT
model only uses station data closest to the centroid of each sub-basin. High-resolution gridded data are converted into abundant virtual stations, but the amount of the sub-basin is difficult to match with the resolution of gridded precipitation. The abundant virtual stations are directly inputted to SWAT model, this leads to inaccurate representation of the sub-basin precipitation input data because most virtual stations are ignored (Tuo et al., 2016). Hence, the current method of representing precipitation in the SWAT model is difficult to reflect the high-resolution superiority of gridded data, a
reasonable method of building virtual stations must be developed.

In this study, we are built virtual precipitation stations for each sub-basin that adopts the mean precipitation of the grid within each subbasin; all grids data can be utilized to build virtual stations on sub-basin scale. Building virtual precipitation station are transferred gridded precipitation data into a sub-basin average and then into SWAT. Thus, the superiority of gridded data in horizontal distribution can be maximized and the representation of sub-basin precipitation input data can be
improved. The virtual precipitation stations are built through the following technical trick: (1) the spatial distribution of the sub-basin was pre-divided, and a grid-sized buffer of the sub-basin boundary was set; (2) the grid precipitation data were converted into point data, whose spatial position is the grid centre; (3) the mean precipitation of all points within the boundary of each sub-basin was calculated and used as values for the virtual precipitation station; and (4) the longitude and



latitude of the sub-basin centroid were used directly as the spatial position of the virtual precipitation station to ensure that each sub-basin reads only one specified station. The virtual station elevation was calculated by the mean elevation of all points within the boundary of the sub-basin.

### 3.4 Precipitation lapse rate

Precipitation significantly varies with elevation because of the relatively complicated terrain in the mountainous region. The SWAT model allows the division of the sub-basin into the elevation bands and sets the precipitation lapse rate to correct the vertical precipitation variability to represent the variability of precipitation caused by elevation changes (Zhang et al., 2015). Previous studies usually regarded the precipitation lapse rate as a calibrated parameter, leading to a low spatial heterogeneity and high precipitation uncertainty. High-resolution gridded precipitation with detailed and accurate

information of vertical distribution necessitates the calculation of precipitation lapse rate on the sub-basin scale. In this study, the lapse rate for each sub-basin is calculated; and the lapse rate serves as input parameters for the SWAT model. The superiority of the gridded precipitation in vertical distribution can be maximised using this method, and the precipitation input parameters for the SWAT model can be optimised.

Several studies used linear regression models to analyse the precipitation variability with elevation in the upper HRB

(Liu et al., 2011; Chen et al., 2014; Chang et al., 2002). Linear regression functions were used to calculate the precipitation lapse rate on the sub-basin and mean annual scale. The linear regression function is shown in Eqs. (2) and (3).

If:

$$P = a - plr \times H \,, \tag{2}$$

then:

$$plr = (P - a)/H \,, \tag{3}$$

where $plr$ is the precipitation lapse rate (mm/km); $P$ is the precipitation at different grids; $a$ is a constant; and $H$ is the elevation at different grids.

## 4 Results

### 4.1 Assessing gridded precipitation

The upper HRB is located in the inland and alpine cold mountainous region, where precipitation is mainly influenced by westerlies and Pacific monsoon. Precipitation in the upper HRB also exhibits a large spatiotemporal variability because of convection in mountainous terrain (Ding et al., 1999). Thus, the quality of gridded precipitation data must be assessed. The gridded precipitation based on the interpolation of gauged stations and RCM has been subjected to cross validation. The results indicate that the gridded precipitation exhibits certain credibility in the HRB; several researchers have employed the

gridded precipitation to analyse the climate and hydrology characteristics in the upper HRB (Yang, et al., 2015; Gao et al., 2015; Qin et al., 2016).

Given the complicated mountainous terrain, the quality of gridded precipitation in the upper HRB was assessed. In this study, gridded precipitation was assessed at both time series accuracy and spatial description capability from 2000 to 2014. Gauged precipitation of the Qilian and Yeniugou meteorological stations was selected to compare the gridded data. These

two stations are located in the east and west tributaries of the basin, which exhibits relatively high representativeness. Precipitation gauged by 13 hydrological stations is also used to evaluate the vertical distribution of precipitation.

### 4.1.1 Time series accuracy

Time series accuracy was evaluated by comparing gauged precipitation with gridded precipitation in a time series to assess the performance of the gridded data. Figs. 3 and 4 show the daily and monthly comparison results in the Qilian and




Yeniugou stations, respectively. In the scatter diagram, the points are symmetrically distributed in both sides of the 1:1 lines, indicating that the gauged data are similar to the gridded data in total value. The point distribution is scattered, and the correlation is relatively high. The comparison of the monthly results indicated that the monthly gridded precipitation is close to the gauged precipitation, and their change trends are consistent.

In this study, determination coefficients ($R^2$), root-mean-square error (RMSE) and percent bias (PBIAS) were used to assess the quality of the gridded precipitation data (Legates et al., 1999; Gupta et al., 1999; Singh et al., 2005).. Table 1 shows the criteria used to evaluate Qilian and Yeniugou. At the yearly scale, the mean annual gridded precipitation is close and slightly lower than the gauged data. The $R^2$ values of Qilian and Yeniugou at the daily scale reached 0.24 and 0.33, respectively. Moreover, the $R^2$ values at the monthly scale all reached 0.99, indicating a strong correlation relationship. The

daily and monthly scales of PBIAS were controlled within ±1% and the RMSE values were approximately 3 mm; the error was relatively low. In summary, precipitation was slightly underestimated by the gridded data, but the gridded precipitation data exhibited satisfactory performance in terms of time series accuracy.

### 4.1.2 Spatial description capability

The superiority of high-resolution gridded precipitation is attributed to its spatial description ability; thus, the spatial

distribution of precipitation must be assessed. Fig. 5 shows the mean annual precipitation distribution of the gridded data. The mean annual gridded precipitation values in the entire basin, the east tributary, the west tributary and the mainstream are 513, 589, 505 and 422 mm, respectively. In the basin, precipitation decreased from the southeast to the northwest, and precipitation from the south face is higher than that from the north face. The maximum precipitation zone occurred in the northern east tributary, and the minimum precipitation zone occurred in the basin outlet. The elevation band of the maximum

precipitation occurred from 4,300 m to 4,800 m. A previous study reported that the location of the maximum precipitation band is related to the degree of dryness and wetness. In the middle of the north face of the Qilian Mountains, the maximum wetness degree elevation is 4,600 m and the maximum precipitation band range from 4,500 m to 4,700 m (Ding et al., 1999; Wang et al., 2009; Li et al., 2013). These conclusions are consistent with the gridded precipitation distribution. However, gridded precipitation was relatively overestimated in high-altitude areas.

The spatial precipitation distribution is strongly correlated with elevation (Fig. 5). Apparently, the mean annual gridded precipitation is high in the high-altitude area and low in the river valley area. Fig. 6 shows the scatter diagram of the mean annual gridded precipitation and its elevation. For the east tributary and mainstream, the $R^2$ values reached 0.89 and 0.74, respectively, indicating apparent strong correlations. The precipitation lapse rate of the east tributary and the mainstream is 165 mm/km, which is close to the precipitation lapse rate from observation data (171 mm/km). Notably, the $R^2$ value of the

west tributary is the lowest (0.16), and its precipitation lapse rate is 84 mm/km. The $R^2$ value of the entire basin is 0.37, and its precipitation lapse rate is 120 mm/km. The east tributary and the mainstream are stronger than the west tributary in terms of precipitation lapse rate. The spatial precipitation lapse rate distribution is consistent with the findings of Chen and Liu (2014, 2011). The precipitation lapse rate of the west tributary is lower than that of the east tributary, mainstream and entire basin. Most of the gauged stations are located at the valley and shallow mountainous areas, with elevations are all lower than

3,500 m. Most stations are near the east tributary and mainstream areas. Thus, the gauge stations lack representativeness for the west tributary. Nevertheless, the spatial precipitation lapse rate distribution has a certain reference value. As a whole, gridded precipitation and elevation exhibited an obvious linear regression relationship; that is, precipitation increases with elevation. In addition, gridded data can be used to describe the horizontal and vertical precipitation distribution in the study area.

### 4.2 Distribution of virtual stations

The spatial discretisation scheme of the SWAT model for precipitation is classified as a lumped type and uses data from the grid closest to the centroid of each sub-basin. The distribution and amounts of the sub-basin can be used to determine





where and how many grids can be entered into the SWAT model. The total number of grid data within the upper HRB reached 1,113 if direct input gridded precipitation will render most grids as ignored. Scale transformation by building virtual precipitation stations is important for grid upscale to sub-basin scale. Thus, the drainage area threshold of the sub-basin division is critical for scale transformation, which determines the distribution and amount of the sub-basin. The setting of drainage area threshold considered regional climate and terrain features, which influence the hydrological processes. The SWAT model can generate a large amount of sub-basin and relatively independent geographical area. Moreover, combining the studies on the optimal drainage area threshold for the upper HRB, the drainage area threshold was set at 50 km² (Lu et al., 2015). The SWAT model generated 97 sub-basins. The centroid of each sub-basin was considered as a virtual precipitation station that adopts the mean precipitation of the grid within each subbasin. From the distribution and amounts of subbasin, 97 virtual precipitation stations of time-series were built to transfer gridded data in to a sub-basin average (Fig. 7). By this method, the SWAT model can reasonably discretize sub-basin precipitation input data. Scale transformation can effectively maximise the superiority of the gridded precipitation in horizontal distribution and optimise the precipitation inputs for the SWAT model.

### 4.3 Distribution of precipitation lapse rate

Precipitation and elevation exhibit a linear regression relationship in the upper HRB; as such, linear regression functions are widely used to compute the precipitation lapse rate (Chen et al., 2014). Linear regression functions of the mean annual gridded precipitation and elevation were established to calculate the precipitation lapse rate at the sub-basin scale. Finally, the precipitation lapse rate of 97 sub-basins was obtained (Fig. 8). A 500 m interval was used to divide the elevation bands for the sub-basin. The precipitation lapse rate and elevation bands were combined to correct the vertical distribution of precipitation, maximising the superiority of the high-resolution gridded precipitation.

Fig. 8 shows the decreasing trend of the precipitation lapse rate from the southeastern to the northwestern areas; the change trends and spatial distribution are consistent with precipitation. The precipitation lapse rate ranges from 40 mm/km to 235 mm/km and decreased from southwest to northwest. The mean precipitation lapse rates are 120 mm/km in the entire basin and 165 mm/km in the east tributary and mainstream. The precipitation lapse rate of the west tributary is 84 mm/km. In summary, the distribution of the precipitation lapse rate is consistent with the previously reported precipitation lapse rate in the upper HRB (Chang et al., 2002; Liu et al., 2011). Thus, a linear regression function of precipitation and elevation is appropriate to calculate the precipitation lapse rates.

### 4.4 Model calibration and performance

The applications of the SWAT model on the large scale and long-term series simulation are mainly concentrated on monthly scale, and most hydrological models are simulated monthly runoff in the upper HRB (Zhao et al., 2005; Jia et al., 2009; Li et al., 2009; Gao et al., 2015). Thus, this study was simulated monthly runoff and evaluated on monthly scale, and the hydrological process was analysed from the monthly simulation results. The SWAT simulation results can be used to compare with similar hydrological models in this study area. The SWAT model was used to simulate the monthly runoff from January 2000 to December 2014. The model was calibrated for the January 2003 to December 2008 period and validated for the January 2009 to December 2014 period; the January 2000 to December 2002 period was regarded as the warm-up period. A comprehensive hydrologic calibration was performed for the upper HRB. The hydrologic calibration followed the multi-temporal, multi-variable, multi-site principles and used the observed data, hydrological characteristics and expert knowledge of the basin to improve the accuracy of the runoff simulation (Xu et al., 2015). Parameter sensitivity was analysed by SWAT-CUP in the upper HRB. The 10 most sensitive parameters in the 3 subregions were manually calibrated and validated. Table 2 shows the most sensitive parameters and their fitted values.

The SWAT model used the observed and simulated monthly runoff statistics at three hydrological stations (Qilian, Zhamashike and Yingluoxia). Three goodness-of-fit measures, i.e. $R^2$, PBIAS and Nash–Sutcliffe efficiencies (NS), are used





(Nash et al., 1970; Legates et al., 1999; Gupta et al., 1999; Singh et al., 2005). The runoff simulated by the SWAT model was compared with the observed runoff at the three hydrological stations. In the hydrographs, the seasonal dynamics of the simulated runoff were well fitted with the observed runoff, except for Qilian and Zhamashike, which exhibited poor performance at the peak value in some years (Fig. 9). Statistical analyses of the calibration and validation periods showed

that the $R^2$ values ranged from 0.76 to 0.93, indicating that the simulation exhibits a strong correlation with the observation findings. The NS ranged from 0.71 to 0.92; thus, the simulation exhibits high credibility. PBIAS values ranged from −14.02% to 11.51%, indicating that the model slightly overestimated the runoff in the calibration period and underestimated the runoff in the validation. However, the PBIAS values are still within a reasonable range. As a whole, model performance in the validation period is better than that in the calibration period.

10       The daily runoff simulation and previous studies on the regional meteorological and hydrological characteristics can also provide reference to validate the SWAT model application better. Considering the daily time step with a high uncertainty, the daily runoff simulated by the SWAT model was compared with the observed runoff at the Yingluoxia station in 2007 and 2013 (Fig. 10). These two years are typical normal flow year and high flow year, which exhibit relatively high representativeness. In the hydrographs, the daily dynamics of the simulated runoff were well fitted with the observed runoff.

The $R^2$ values are higher than 0.60, the NS are higher than 0.55 and the PBIAS are controlled within ±10%. It's indicating that the model exhibit satisfied performances on the daily scale. From the model simulation results, the base flow coefficient is 0.46, which is close to the result of the base flow separation (0.44) (Zhang et al., 2011). Snowmelt runoff simulated by the model, which is lower than the studies of Wang et al. (2006; 2010). The hydrographs show that the model performed well during the snowmelt period (April to May). The hydrographs also indicated that snowmelt runoff simulation is reasonable.

The actual evapotranspiration is 318 mm, which is close to the remote sensing data (306 mm) (Wu et al., 2012). The potential evapotranspiration estimated by the SWAT model is 575.3 mm, which is relatively higher than the HBV runoff model simulation (500.4 mm) (Kang et al., 1999).

       In summary, the simulation results exhibit good and very good performances that satisfied the accuracy and reliability requirements of the SWAT model (Moriasi et al., 2007). Simulation with the SWAT model used high-resolution gridded precipitation and comprised building virtual stations and calculating lapse rates to optimise the precipitation input parameters

and improve the accuracy of the results. After using high-resolution gridded precipitation, the model can simulate accurate and detailed spatial distribution of water balance components, thereby improving our understanding of the regional hydrological processes.

### 4.5 Water balance component characteristics

30       The water balance components considered in this study include precipitation (PREC), evapotranspiration (ET), water yield (WYLD) and soil water content (SW). Table 3 shows the mean annual values of the water balance components from 2003 to 2014 in different regions. For the entire basin, precipitation, evapotranspiration and water yield are 525.5, 318.1 and 194.4 mm, respectively. This finding indicates that the water balance components were relatively balanced. The mean annual precipitation is close to the original gridded precipitation (513 mm), indicating that the scale transformation and precipitation

lapse rate calculation are reasonable. Evapotranspiration is similar to the remote sensing data (306 mm) (Wu et al., 2012). The differences of water balance components in different regions were determined by precipitation. The runoff coefficients in different regions are similar, and the coefficient of the entire basin is 0.37.

### 4.5.1 Spatial variability of water balance components at the landscape scale

       Fig. 11 shows the percentage, runoff contribution and runoff coefficient of water balance components on the landscape

scale. The meadow is the dominant vegetation type, which accounts for 43.8% of the basin area and contributes 44.8% of the runoff. The runoff coefficient of the meadow is 0.37, which is equal to that of the entire basin. The area of sparse vegetation accounts for 19.6% of the basin area and contributes 23.6% of the runoff. The runoff coefficient of sparse vegetation is the



highest (0.42) at all of the landscapes. The sparse vegetation is generally distributed at high-altitude areas with low temperatures, high precipitation and alpine cold desert. Thus, evapotranspiration and soil water content in this area are relatively low, and the runoff coefficient is relatively high. The shrub accounts for 16.8% of the basin area and contributes to 17.2% of the runoff. The meadow, sparse vegetation and shrub account for 79.4% of the basin area and contribute 85.6% of the runoff, which are mainly water yield landscapes. Steppe and coniferous forest account for 17.2% of the basin area and contribute 12.2% of the runoff; however, their runoff coefficients are low because of the interception of the canopy and roots. Thus far, whether the forests in the alpine cold mountain region generate runoff remains unclear. The forest in the study area includes coniferous forest and shrub, which contribute 4.5% and 17.2% of the total runoff, respectively. These results are similar to the findings of the small catchment experiment and hydrological simulation in the upper HRB (He et al., 2012; Yin et al. 2016). The water body includes river, snow and glacier, and the runoff coefficient is relatively high. The desert and crop areas are lower and contribute only 0.6% of the total runoff.

### 4.5.2 Spatial variability of water balance components at the elevation band scale

Elevation significantly affects the hydrological processes in alpine cold mountainous regions. According to the vertical distribution of vegetation, the basin was divided into five elevation bands, namely, 1,667–2,800, 2,800–3,500, 3,500–4,000, 4,000–4,500 and 4,500–5,008 m. Fig. 12 shows the percentage, runoff contribution and runoff coefficient of water balance components on the elevation band scale. The largest area is the elevation band at 3,500–4,000 m, which accounts for 40.1% of the basin area and contributes 42.2% of the runoff. The runoff coefficient is 0.38, which is close to the runoff coefficient of the entire basin. The elevation band at 4,000–4,500 m accounts for 25.5% of the basin area and contributes 29.3% of the runoff. This area is featured as cold and wet; thus, the evapotranspiration percentage and soil water content are relatively low, and the runoff coefficient is relatively high. The elevation band at 3,500–4,500 m accounts for 65.6% of the basin area and contributes 71.5% of the total runoff. Thus, the basin runoff is mainly derived from the high-altitude regions. The elevation band at 2,800–3,500 m is characterised as warm and dry; thus, precipitation is mainly consumed by evapotranspiration and stored in soil, and the runoff coefficient is relatively low. The elevation band at 4,500–5,008 m contains a large area of snow and glacier, which exhibits the highest runoff coefficient (0.44) because of snow melting. The elevation band at 1,667–2,800 m is desert and steppe, and all of the water balance components are low. Overall, climate variability with elevation significantly affects the distribution of water balance components.

### 4.5.3 Spatial variability of water balance components at the sub-basin scale

Changes in precipitation are the dominant factor that induces changes in water balance components (Zhang et al. 2016). Fig. 12 shows the spatial variability of the mean annual value of water balance components on sub-basin scale. Precipitation over the basin ranges from 231 mm to 670 mm and decreases from the southeast to the northwest. Precipitation of the east tributary is higher than that of the west tributary, and that of the mainstream is the lowest. Evapotranspiration ranges from 220 mm to 560 mm, and the mean value of the entire basin is 306 mm. The water yield ranges from 13 mm to 376 mm and is less than 60 mm in the basin outlet covered by desert. Thus, the underlying surface significantly affects the water yield capacity of the sub-basin. Evapotranspiration exhibits distributions that are similar to that of precipitation. However, the pattern of soil water content varies. Affected by elevation and landscape, the soil water distribution of the east tributary is similar to that of the west tributary, and that of the mainstream is low. The soil water content ranges from 5 mm to 150 mm, and the average value of the entire basin is 52 mm.

Fig. 13 shows the mean annual change trends of water balance components on the sub-basin scale from 2003 to 2014. The precipitation change rate in the entire basin is 0.18 mm/a, which indicates a slightly increasing trend. The precipitation increase rate decreased from the southwest to the northwest and varied from −7.4 mm/a to 3.55 mm/a. The large precipitation increase rate was concentrated in the western basin; only four sub-basins presented a decreasing trend. The increasing rate of evapotranspiration decreased from the southeast to the northwest, ranging from –2.51 mm/a to 3.6 mm/a,





with an average value of 0.78 mm/a. The water yield decreased in the entire basin (−1.29 mm/a), ranging from −11.75 mm/a to 2.93 mm/a. The spatial distribution of water yield change rate is consistent with that of precipitation; the increasing trend in the western basin is more obvious compared with that in the other regions. The soil water content decreased in the entire basin (−1.18 mm/a), ranging from −3.35 mm/a to 0.46 mm/a at the sub-basin scale.

## 5 Discussion

This study mainly aims to optimise the input parameters for hydrological simulation using high-resolution gridded precipitation and provide reference for water resource assessment and management in data-scarce regions. The SWAT model uses gridded precipitation data from the grid closest to the centre of each sub-basin and is corrected by the elevation band and lapse rate. Scale transformation is proposed by building virtual precipitation stations and calculating the precipitation lapse rate; this results in the gridded data that is upscale from the grid to sub-basin scale. To some extent, these methods can be used to optimise the precipitation input parameters for the SWAT model effectively and maximise the horizontal and vertical distribution superiorities of the high-resolution gridded precipitation. However, the 1,113 grids were converted into 97 virtual stations, which simplified the spatial distribution of precipitation. Thus, further studies should focus on the optimal drainage threshold area of the sub-basin division. Based on basin climate and terrain, the division into the sub-basin with a large amount, and then the building of virtual station with a high density is necessary. Previous studies showed that precipitation and elevation may be best described by log-linear or exponential functions (Daly et al., 1994). In the present study, liner regression functions were selected because the precipitation lapse rate is considered the mean annual value on the sub-basin scale in the SWAT model. Although this method simplified the vertical variability of precipitation with elevation, a linear regression function is suitable for calculating the precipitation lapse rate for the SWAT model.

The SWAT model achieved a good monthly runoff simulation on the large scale and long-term series, which is sufficient to support the study on the water balance component characteristics on the mean annual scale. This result can provide a credibility reference for basin water resource assessment and management. And most hydrological models are simulated monthly runoff in this study area, our research can be used to compare with the previous study. However, the monthly simulation hardly reflects the time series superiority of the daily gridded precipitation. Thus, the further study on the daily simulation and analysis of water balance component characteristics on the inter-annual and small catchment scale is necessary.

The hydrological model is widely used in the upper HRB to study hydrological processes. Compared with previous studies, the simulation accuracy derived in the present study has yet to be improved (Li et al. 2009, 2010, 2011; Lu et al. 2015). For model climate forcing, only precipitation inputs used high-resolution gridded data; the temperature, wind speed, solar ration and relative humidity still used gauged data, which were scarce and unevenly distributed. The high-resolution gridded data of other climate elements should be applied in the SWAT model. For validation data, the precipitation lapse rate, soil water content and evapotranspiration lack gauged data. These factors influence the accuracy of the model simulation. The runoff coefficient of the coniferous forest is relatively higher than those reported in previous studies; improvement of the calibration model is necessary (Gao et al., 2015; He et al., 2012; Yin et al., 2016). Future studies should focus on these limitations to investigate the SWAT model driven by high-resolution gridded data and to improve the model performance.

## 6 Conclusion

High-resolution gridded precipitation data were subjected to quality assessment and selected as forcing data for SWAT model hydrological simulation in the upper HRB. Scale transformation was proposed by building virtual precipitation and calculating precipitation lapse rate on the sub-basin scale to optimise the input parameters for the SWAT model. The monthly runoff was simulated from 2000 to 2014 and validated using the observed data. The spatial variability of the water balance components was analysed based on the simulation, which is used to provide references for the assessment and management of the basin water resources in data scarce regions. The main conclusions are presented as follows:

(1) The time series accuracy of gridded precipitation were assessed in Qilian and Yeniugou. The RMSE is approximately 3




mm and the PBIAS is controlled within ±1%; both exhibit a strong correlation. The spatial distribution of gridded precipitation decreased from the southeast to the northwest and increased with elevation, which consisted of real precipitation features. Thus, these datasets exhibited high time series accuracy and spatial description ability in the study area.

(2) The scale transformation of gridded precipitation was conducted by building virtual stations and calculating lapse rates. The 97 virtual stations were built with the mean precipitation of all grids within each sub-basin. The precipitation lapse rate ranges from 40 mm/km to 235 mm/km and decreased from the southeast to the northwest. The mean precipitation lapse rates are 120 mm/km in the entire basin and 165 mm/km in the east tributary and mainstream. The precipitation lapse rate of the west tributary is 84 mm/km. Through this method, the superiority of gridded precipitation in the

horizontal and vertical distributions is maximised, and the input precipitation parameters of the SWAT model are optimised.

   (3) From the accurate representation of the sub-basin precipitation input data, the SWAT model exhibited a good monthly runoff simulation compared with the observed data in Qilian, Yeniugou and Yingluoxia stations. Statistical analyses of the calibration and validation periods showed that the $R^2$ value was higher than 0.71, the NS was higher than 0.76 and

the PBIAS was controlled within ±15%. The model performance in the validation period is better than that in the calibration period. The base flow coefficient, snow melt runoff and potential evapotranspiration simulated by the model are consistent with those of previous studies.

   (4) The spatial variability of water balance components was analysed on sub-basin, elevation band and landscape scales. The meadow and sparse vegetation are major water yield vegetation types, occupying 63.4% of the basin area and

contributing 68.3% of the runoff, with a runoff coefficient reaching 0.39. The band at 3,500 m to 4,500 m is the major water yield elevation band, occupying 65.5% of the basin area and contributing 71.5% of the runoff, with a runoff coefficient reaching 0.39. At the sub-basin scale, the spatial distribution of the water yield and evapotranspiration are consistent with that of precipitation, decreasing from the southeastern to the northwestern areas. However, the spatial distribution of soil water content is similar to that of the western and eastern areas because of the landscape and

elevation band effect. In the terms of the entire basin, the precipitation and evapotranspiration presented a slightly increasing trend, whereas the water yield and soil water content presented a slightly decreasing trend.

**Acknowledgments:** This work was supported by the National Natural Science Foundation of China (41571031, 91225302, 41601038, 41601036). The authors are grateful to the Cold and Arid Regions Science Data Centre at Lanzhou (http://westdc.westgis.ac.cn) for data support. The authors thank the editor for constructive comments that have significantly

improved this work.

**Author Contributions:** Songbing Zou designed the research. Hongwei Ruan performed modelling analysis and wrote the manuscript. Zhentao Cong and Yuhan Wang provided gridded precipitation data. Zhenliang Yin, Zhixiang Lu, Fang Li and Baorong Xu made editing corrections and improvements to the manuscript.

**Conflicts of Interest:** The authors declare no conflict of interest.

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




**Table captions**

Table 1. Calculation of evaluation criteria.

Table 2. Most sensitive parameters.

Table 3. Water balance components for different regions.

**Table 1.** Calculation of the evaluation criteria.

| Station | Yearly scale | | Daily scale | | | Monthly scale | | |
|---|---|---|---|---|---|---|---|---|
| | Gauge data (mm) | Gridded data (mm) | $R^2$ | RMSE (mm) | PBIAS (%) | $R^2$ | RMSE (mm) | PBIAS (%) |
| Qilian | 424.5 | 422.8 | 0.24 | 3.27 | 0.62 | 0.99 | 2.73 | 0.62 |
| Yeniugou | 460.4 | 459.0 | 0.33 | 3.04 | 0.30 | 0.99 | 2.57 | 0.30 |

**Table 2.** Most sensitive parameters.

| Parameter | Description | Range | Value | Sensitive |
|---|---|---|---|---|
| Ch_K2 | Effective hydraulic conductivity in main channel alluvium (mm/h) | 0–500 | 8–15 | 1 |
| Cn2 | Initial SCS runoff curve number for moisture condition II | 35–98 | 43–95 | 2 |
| Plaps | Precipitation lapse rate (mm/km) | −1000–1000 | 40–235 | 3 |
| Esco | Soil evaporation compensation factor | 0–1 | 0.83–0.90 | 4 |
| Alpha_Bf | Base flow alpha factor (days) | 0–1 | 0.06–0.072 | 5 |
| Smfmn | Melt factor on December 21 (mm $H_2O$/°C day) | 0–20 | 1 | 6 |
| Sol_Awc | Available water capacity of the soil layer (mm $H_2O$/mm soil) | 0–1 | 0.1–0.22 | 7 |
| Tlaps | Temperature lapse rate (°C/km) | −10–10 | −5 | 8 |
| Gw_Delay | Groundwater delay time (days) | 0–500 | 31 | 9 |
| Smfmx | Melt factor on 21 June (mm $H_2O$/°C day) | 0–20 | 2.5 | 10 |

**Table 3.** Water balance components for different regions.

| Region | Area (km$^2$) | PREC (mm) | ET (mm) | WYLD (mm) | SW (mm) | Runoff coefficient | Runoff contribution (%) |
|---|---|---|---|---|---|---|---|
| East tributary | 2,504 | 609.8 | 364.4 | 229.9 | 63.5 | 0.37 | 29 |
| West tributary | 5,032 | 522.8 | 310.3 | 199.5 | 58.5 | 0.38 | 52 |
| Main stream | 2,482 | 446.2 | 287.4 | 148.3 | 27.7 | 0.33 | 19 |
| Entire basin | 10,018 | 525.5 | 318.1 | 194.4 | 52.1 | 0.37 | 100 |



**Figure captions**

Figure 1. Upper Heihe River Basin.

Figure 2. Vegetation map of the upper Heihe River Basin.

Figure 3. Scatter diagram of daily gauge precipitation and daily gridded precipitation in Qilian (a) and Yeniugou (b) (Red
line is trend line, and blue line is 1:1 line).

Figure 4. Comparison of monthly gauge precipitation and gridded precipitation.

Figure 5. Distribution of average annual gridded precipitation.

Figure 6. Scatter diagram of gridded precipitation and its elevation in east tributary (a), west tributary (b), mainstream (c),
entire basin (d) and observations (e).

Figure 7. Distribution of virtual precipitation station.

Figure 8. Distribution of subbasin precipitation lapse rate.

Figure 9. Monthly observation and simulated runoff of Qilian (a), Zhamashike (b) and Yingluoxia (c).

Figure 10. Daily observation and simulated runoff on Yingluoxia in 2007 (a) and 2013 (b)

Figure 11. Spatial variability of water balance components at the landscape scale and their runoff coefficients.

Figure 12. Spatial variability of water balance components at the elevation band scale and runoff coefficient

Figure 13. Annual average value of water balance components at the subbasin scale.

Figure 14. Change trend of water balance components at the subbasin scale.

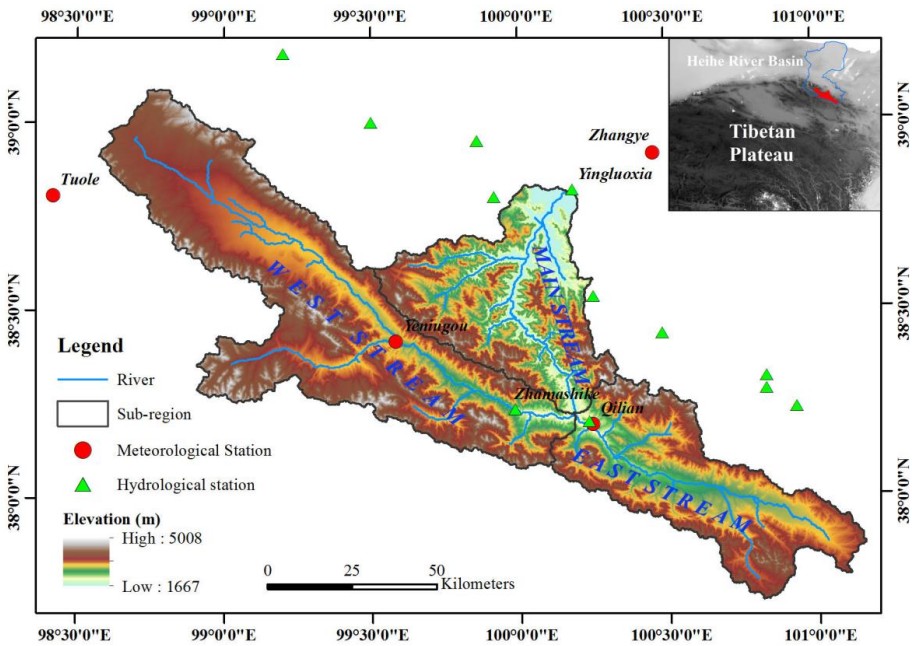

**Figure 1.** Upper Heihe River Basin.





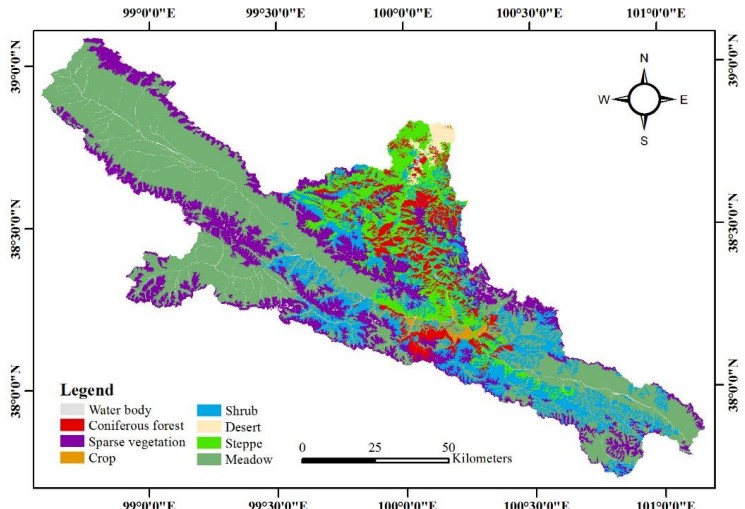

**Figure 2.** Vegetation map of the upper Heihe River Basin.

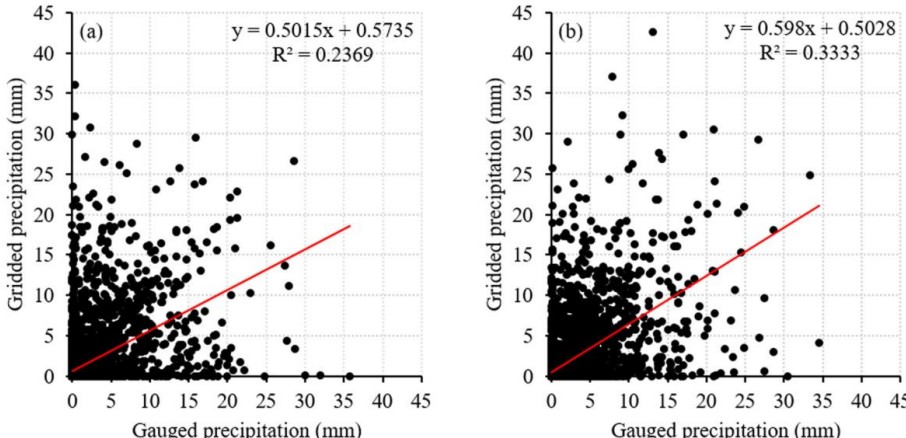

5      **Figure 3.** Scatter diagram of daily gauge precipitation and daily gridded precipitation in Qilian (a) and Yeniugou (b) (red line, trend line; blue line, 1:1 line)

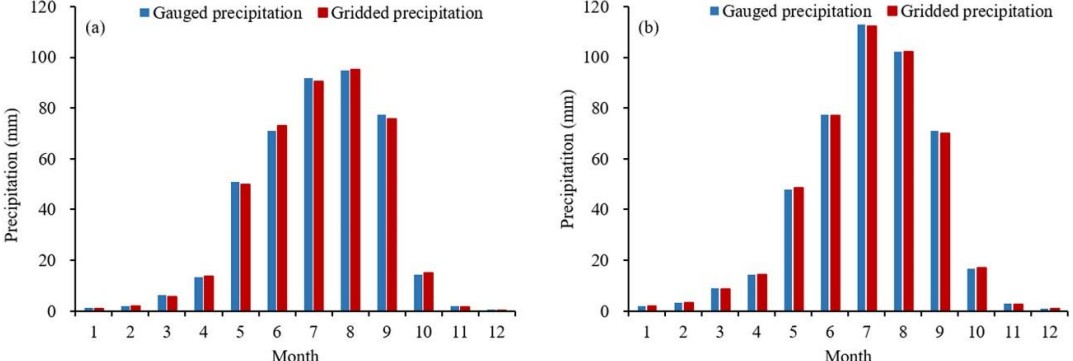

**Figure 4.** Comparison of monthly gauge precipitation and gridded precipitation.





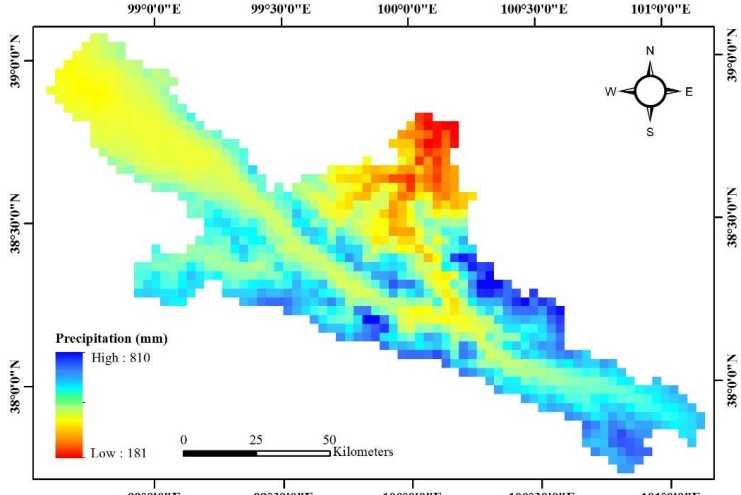

**Figure 5.** Distribution of the average annual gridded precipitation.





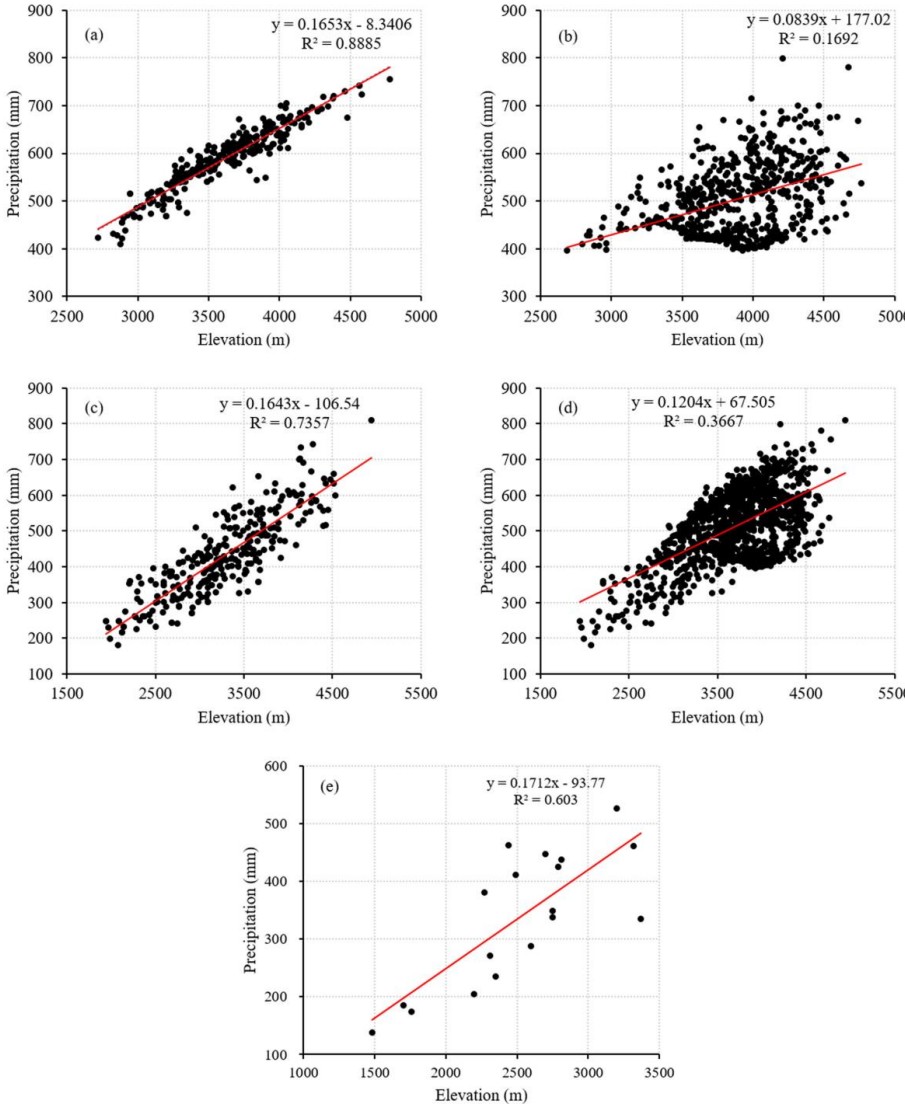

**Figure 6.** Scatter diagram of the gridded precipitation and its elevation in the east tributary (a), west tributary (b), mainstream (c), entire basin (d) and observations (e).





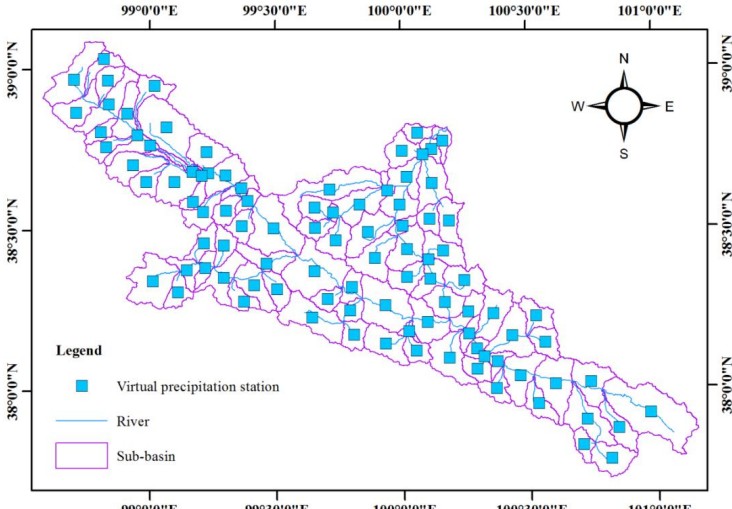

**Figure 7.** Distribution of the virtual precipitation station.

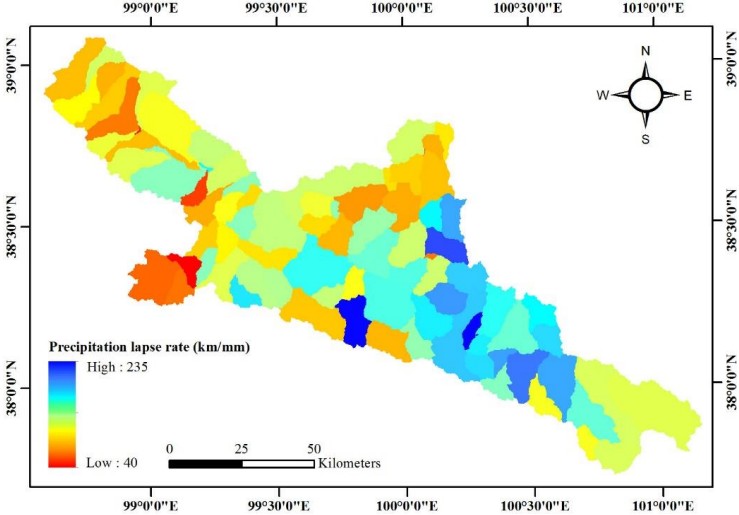

5            **Figure 8.** Distribution of the sub-basin precipitation lapse rate.





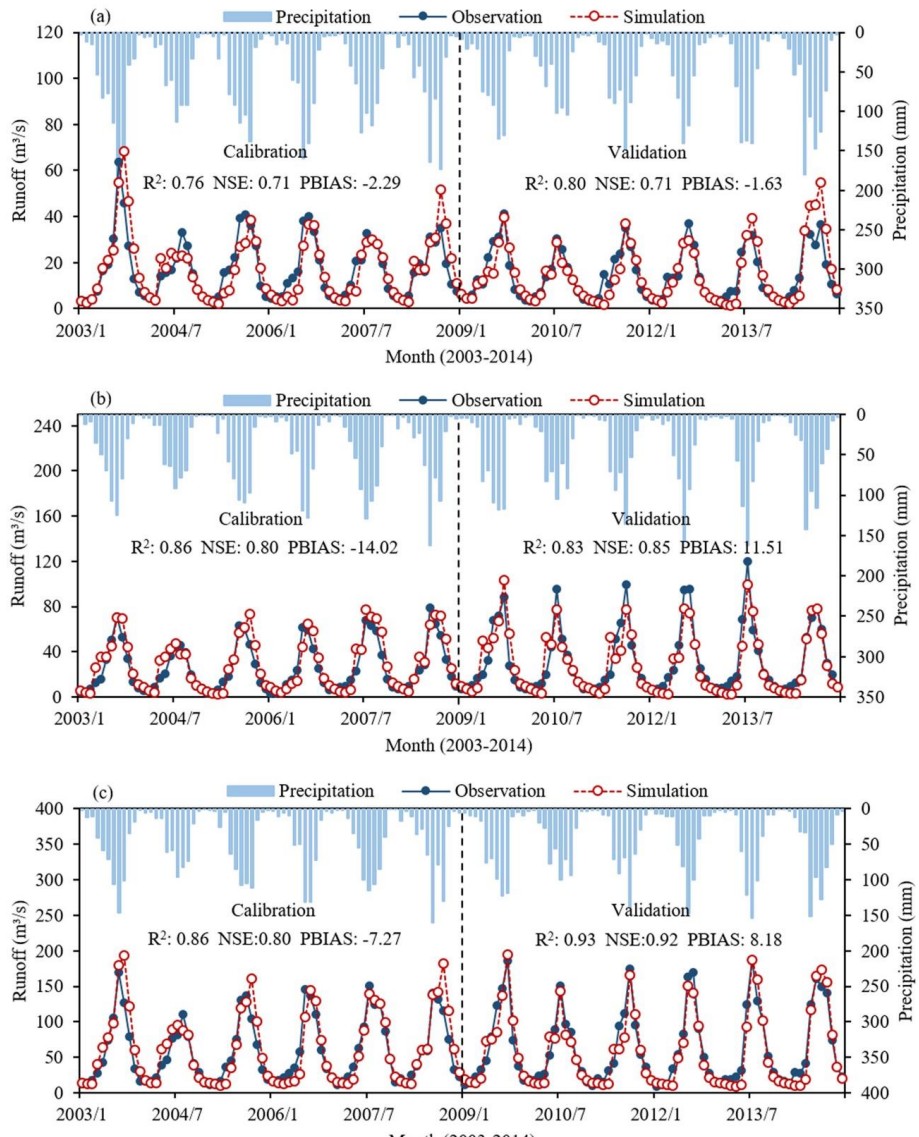

**Figure 9.** Monthly observation and simulated runoff of Qilian (a), Zhamashike (b) and Yingluoxia (c).





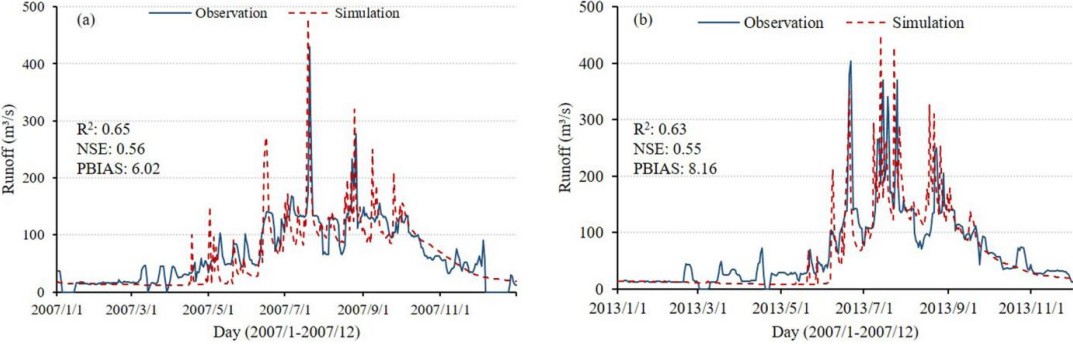

**Figure 10.** Daily observation and simulated runoff of Yingluoxia in 2007 (a) and 2013 (b)

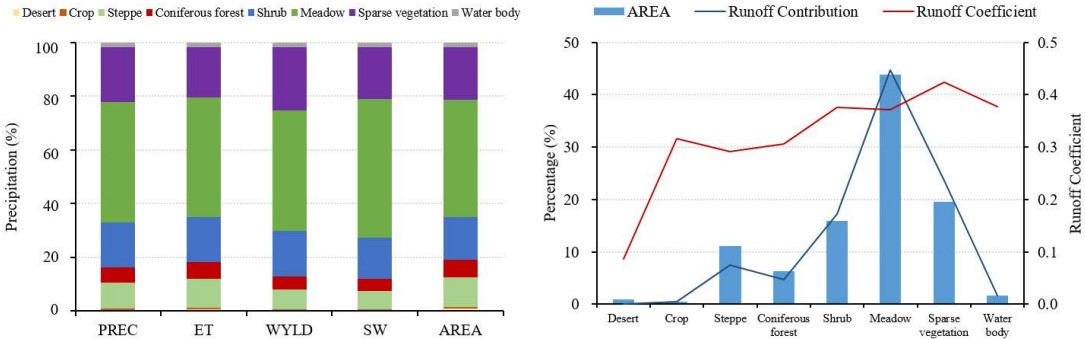

5           **Figure 11.** Spatial variability of water balance components at the landscape scale and their runoff coefficients.

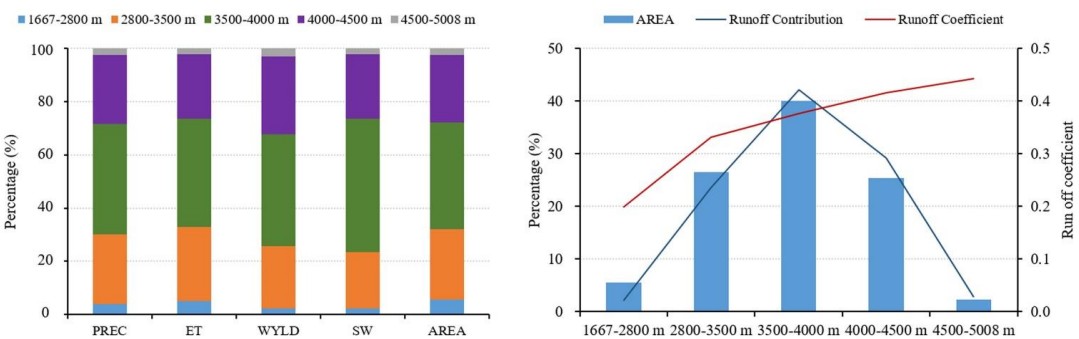

**Figure 12.** Spatial variability of water balance components at the elevation band scale and runoff coefficient



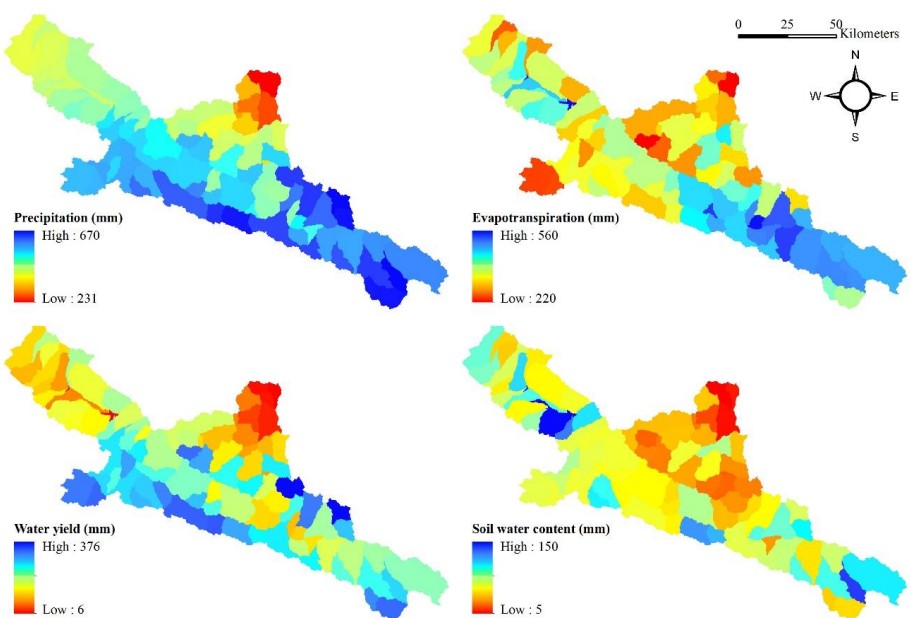

**Figure 13.** Annual average value of the water balance components at the sub-basin scale.

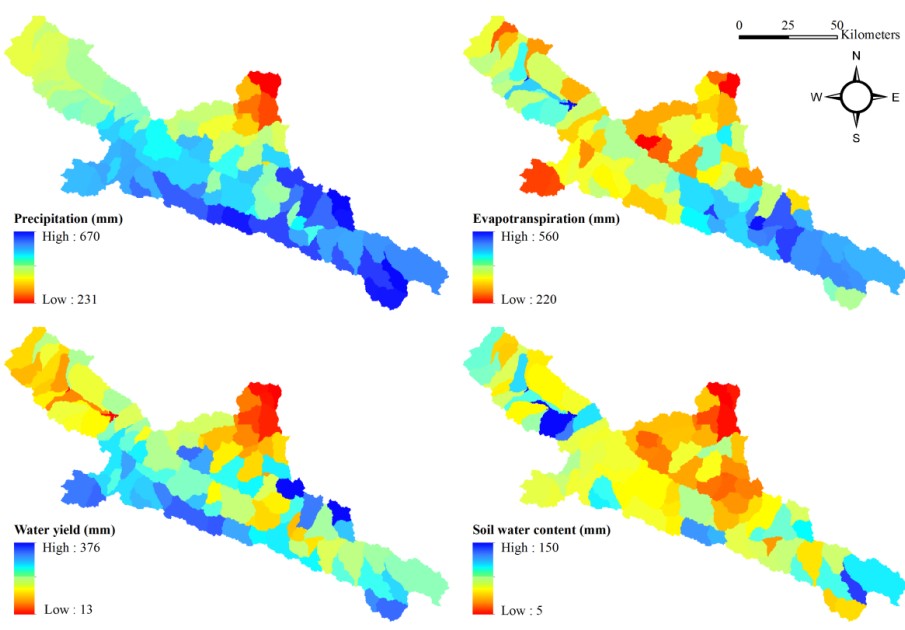

**Figure 14.** Change trend of the water balance components at the sub-basin scale.