# Peer review of "Runoff simulation by SWAT model using high-resolution gridded precipitation in the upper Heihe River Basin, Northeastern Tibetan Plateau"

_Hydrology and Earth System Sciences, 2016_

## Referee Comment (RC1) · Anonymous Referee #1 · 26 Jan 2017

General comments: This manuscript presents a case study of hydrological simulation using SWAT model in the upper Heihe River in northwestern China. For applying the distributed hydrological model in a gauge-sparse catchment, this study used a gridded precipitation data interpolated from the gauged stations and the regional climate model. The main contents of this paper include validation of the gridded precipitation data, application and validation of the SWAT model, and spatial characteristics of water balance in the upper Heihe basin. The topic discussed in this paper and the major results obtained by this study are interesting. However there are several major issues should be addressed, structure and grammar of this manuscript should be improved

before publication. Specific comments: 1) P2, L12-14: The following reference should be added here. Wang YH, Yang HB, et al. Spatial Interpolation of Daily Precipitation in a High Mountainous Watershed based on Gauge Observations and a Regional Climate Model Simulation. Journal of Hydrometeorology, 2016, DOI: 10.1175/JHM-D-16-0089.1 2) P2: Author should need to explain why select SWAT model in this study. What are the major advantages of SWAT comparing with many other distributed hydrological models? 3) P3: How are the glaciers in the study area? Do you consider the glacier melting runoff in the hydrological simulation? 4) P3: "Gridded precipitation data with daily resolutions of 3 km were used as precipitation forcing data downloaded from the Heihe Plan Science Data Centre (HPSD)." Please specify the original source or reference. 5) P4: the gridded precipitation data: please refer the following paper. Wang YH, Yang HB, et al. Spatial Interpolation of Daily Precipitation in a High Mountainous Watershed based on Gauge Observations and a Regional Climate Model Simulation. Journal of Hydrometeorology, 2016, DOI: 10.1175/JHM-D-16-0089.1 6) P 5, Results: The general introduction about the gridded precipitation data should be moved to the Introduction section. The result section should contain the result mainly. 7) P8, Water balance component: Is this a long-term mean water balance at annual scale? Please specify the simulation period. The units of water balance components in Table should be mm/year for precipitation, evapotranspiration and runoff. Please chek the units carefully. 8) Section 4.5.1 and 4.5.2: You mentioned "The landscape follows a distinct vertical zonation and comprises the desert, steppe, shrub, coniferous forest, meadow, sparse vegetation, snow and glaciers" (P3). So what are the differences of results in the two sections? 9) P10: Uncertainty of the hydrological simulation should be discussed.

---

## Referee Comment (RC2) · Anonymous Referee #2 · 8 Feb 2017

The manuscript "Runoff simulation by SWAT model using high-resolution gridded precipitation in the upper Heihe River Basin, Northeastern Tibetan Plateau" by Ruan et al. is about the application of a SWAT model for the respective basin. I am not happy with the manuscript as it lacks of some important evidences about major conclusions. The applied method, to derive the precipitation data may be sufficient, but in the context of SWAT I could not find any comparison between the "past" and "present", how the data set really improves the SWAT modelling. Further more questions can be found in my detailed comments.

Comments:

P1L12 "in inland river basins" is to general, add something like 'Tibet'

P1L15 What kind of RCM is used, maybe better use "a" than "the", as there are many RCM out there.

P1L18 "upscale(d)"?

P1L24 I expected at monthly scale larger NSEs

P2L10 Again what kind of RCM?, better "a" than "the"

P2L12 "depict"?

P2L12 "A Soil and..", this sentence sounds like SWAT can only be applied in this basin

 $\mathsf{P2L45}$  "change trend" sounds strange to me , what is meant by a changing trend: the change of a change is no change?

P3L31 Your data is good? No Outliers? Did you check anything?

P4L8 At this point I asked myself about calibration and validation, you gave some answers to that later in chapter 4.4

P4L12 "Shepard interpolation" I had to google that, better known as "Inverse distance weighting"

P4L13 where these station coming from, you said in chapter 2.2. that you only have 4 met. Station.

P4L15 "RCM was calibrated" How did you do this? Normally there are precipitation parametrisation schemes in RCMs, which more or less adequate model convective processes.

P4L17 "optimal interpolation" who decided how what is optimal?

P4L18 So at the end you did a residual correction, which means to me that at a pixel,

which has a station in it, the observed value is matched by the simulated one? Am I right? Some kind of unbiasedness.

P5L21 "a" is the offset at 0m above ground, as precipitation will not be 0mm at this altitude

P5L27 and L32 "Thus,.." and "Given the.." is unnecessarily doubling of sentences

P5L28 Was this cross validation done by you? Did you really ensure that the two stations have not been used to interpolate the precipitation grids?

P5L38 How did you compare the data? One pixel to one station? Or a mean of 3x3 pixels?

P6L2 "relatively high correlation" I cannot agree with that, there is no correlation at all. Your regression line is defined only by 0mm values, as they are obviously not excluded by this analysis. Tab1. The RMSE for daily precipitation is very large compared to the annual precipitation!!

P6L8 These are not the real correlations, please exclude 0mm values.

P6L24 "was relatively overestimated" by your data? To which reference?

P7L30 " most hydrological models (have)"

P7L31 "Thus, this ..: " sounds not correct

P7L36ff this sentence is not necessary, as I expected that from you

P7L39 Why didn't you apply some auto calibration tool?

P8L13 "typical normal year..." that sounds strange to me

P8L26 "improve the accuracy.." I did not see any proof of that, you did not compare anything only one simulation and a observed time series. That is one of the main problems of your study I cannot judge your results to a former results as there is no comparison.

СЗ

P8L33 "This finding..." that is the character of a balance, I did not expect something different, maybe you skip that.

P9L38 "mean annual change trends" what is meant by that. FIGURE 13 is unclear to me, what is shown there. How do you define a change trend?

P10L13 "Thus,.." I could not follow you logic how is the simple precipitation connected to the drainage threshold?

P10L24 "superiority.." I couldn't see such a superiority of your daily data set! That was not shown in the manuscript sorry!

P10L29 "The temperature..." The RCM should deliver these variables to, they are even more reliable than precipitation from RCMs, so why didn't you use them?

P11L3 and L9 " exhibits high time series accuracy" and "superiority" that was not shown!

P11L25 "trends" In terms of climatology the 14 years you investigated are too short for a trend analysis. You need at least >30a as your observed trends maybe only caused by natural climate variability.

---

## Author Comment (AC1) · 20 Mar 2017

Thank you for your comments and suggestions. Those comments are all valuable and very helpful for revising and improving our paper, as well as important for further study of to our researches. We have studied comments carefully and have made correction which we hope meet with approval. The main corrections in the paper and the responds to comments are as flowing:

1) P2, L12-14: The following reference should be added here. Wang YH, Yang HB, et al. Spatial Interpolation of Daily Precipitation in a High Mountainous Watershed

based on Gauge Observations and a Regional Climate Model Simulation. Journal of Hydrometeorology, 2016, DOI: 10.1175/JHM-D-16-0089.1 Answer: Thus, the gridded precipitation data based on the spatial interpolation of abundant gauged stations and RCM simulation can well depict its spatial heterogeneity, which is suitable for driving the hydrological model (Wang et al., 2017).

2) P2: Author should need to explain why select SWAT model in this study. What are the major advantages of SWAT comparing with many other distributed hydrological models? Answer: Soil and water assessment tool (SWAT) is a physical, semi-distributed hydrological model, which has some advantages in predicting climate change effects on water-related and hydrological processes over a continuous-time (Arnold et al., 2012).

3) P3: How are the glaciers in the study area? Do you consider the glacier melting runoff in the hydrological simulation? Answer: In this study, we did not consider the glacier melting runoff. The SWAT model lack the component of glacier melting runoff, but we simulated snow melting runoff and rainfall runoff on the glacier. Considering the glacier area and runoff contribution are low, and the glacier area changed little in recent years. Thus, ignoring the glacier melting runoff have little influence on the results. We discussed this question in Section 5. Revised in Section 2.1: The glacier area is approximately 34.8 km2, which accounts for 0.35% of the basin area and contributes 3% of the runoff (Kang et al., 1999).

4) P3: "Gridded precipitation data with daily resolutions of 3 km were used as precipitation forcing data downloaded from the Heihe Plan Science Data Centre (HPSD)." Please specify the original source or reference. Answer: Gridded precipitation data with daily resolutions of 3 km were used as precipitation forcing data downloaded from the Heihe Plan Science Data Centre (HPSD), which developed by Wang et al. (2017).

5) P4: the gridded precipitation data: please refer the following paper. Wang YH, Yang HB, et al. Spatial Interpolation of Daily Precipitation in a High Mountainous Watershed based on Gauge Observations and a Regional Climate Model Simulation. Journal of

Hydrometeorology, 2016, DOI: 10.1175/JHM-D-16-0089.1 Answer: The gridded precipitation data with daily resolutions of 3 km for a time series were generated over the HRB (Yang, et al., 2015; Gao et al., 2015; Qin et al., 2016; Wang et al., 2017).

6) P5, Results: The general introduction about the gridded precipitation data should be moved to the Introduction section. The result section should contain the result mainly. Answer: We have removed this paragraph.

7) P8, Water balance component: Is this a long-term mean water balance at annual scale? Please specify the simulation period. The units of water balance components in Table should be mm/year for precipitation, evapotranspiration and runoff. Please check the units carefully. Answer: I'm so sorry for my careless, we have made correction according to your advice. Revised in section 4.5: The water balance components considered in this study include precipitation (PREC), evapotranspiration (ET), water yield (WYLD) and soil water content (SW), which is a long-term mean value at annual scale during the period of 2003-2014. Table 3 shows the mean annual values of the water balance components from 2003 to 2014 in different regions. For the entire basin, precipitation, evapotranspiration and water yield are 525.5, 318.1 and 194.4 mm/a, respectively. It indicates that the water balance components were relatively balanced. The mean annual precipitation is close to the original gridded precipitation (513 mm/a), indicating that the scale transformation and precipitation lapse rate calculation are reasonable. Evapotranspiration is similar to the remote sensing data (306 mm/a) (Wu et al., 2012). The differences of water balance components in different regions were determined by precipitation. The runoff coefficients in different regions are similar, and the coefficient of the entire basin is 0.37.

8) Section 4.5.1 and 4.5.2: You mentioned "The landscape follows a distinct vertical zonation and comprises the desert, steppe, shrub, coniferous forest, meadow, sparse vegetation, snow and glaciers" (P3). So what are the differences of results in the two sections? Answer: On the whole, the landscape follows a distinct vertical zonation. The distribution of landscape with a certain discontinuity and crumbliness, which may

СЗ

be crossed several elevation bands. The area of various landscape types have great differences, and the same landscape in different elevation band has great difference. On contrary, the elevation band has obvious boundaries that may be comprised different landscape. Thus, the water balance features of landscape scale and elevation band scale are obviously different. It is necessary to analyze the spatial variability of water balance components, respectively. Revised in section 4.5.2: Elevation significantly affects the hydrological processes in alpine cold mountainous regions. Although the landscape follows a distinct a vertical elevation band, but the elevation band has obvious boundaries that may be comprised different landscape. There are obvious difference between of them. Thus, it is necessary to analyze the spatial variability of water balance components on elevation band scale.

9) P10: Uncertainty of the hydrological simulation should be discussed. Answer: We discussed uncertainty of the hydrological simulation in Section 5. Revised as follows: To some extent, these methods can be used to optimise the precipitation input parameters for the SWAT model effectively and maximise the horizontal and vertical distribution superiorities of the high-resolution gridded precipitation. However, the 1,113 grids were converted into 97 virtual stations, which simplified the spatial distribution of precipitation. Thus, further studies should focus on the optimal drainage threshold area of the sub-basin division. Based on basin climate and terrain, the division into the sub-basin with a large amount, and then the building of virtual station with a high density is necessary. Previous studies showed that precipitation and elevation may be best described by log-linear or exponential functions (Daly et al., 1994). In the present study, liner regression functions were selected because the precipitation lapse rate is considered the mean annual value on the sub-basin scale in the SWAT model. Although this method simplified the vertical variability of precipitation with elevation, a linear regression function is suitable for calculating the precipitation lapse rate for the SWAT model. The upper HRB is a typical high cold mountainous region. The process of glacier and permafrost are ignored by SWAT model. Considering the glacier area and runoff contribution are low and the glacier area changed little in recent years (Guo

et al., 2014), ignoring the glacier melting runoff have little influence on the results. After the analysis of parameter sensitivity by SWAT-CUP, the 10 most sensitive parameters are achieved. The range of parameter calibration was controlled within in  $\pm 20\%$ . The precipitation event has great uncertainty and randomness, and gridded data have a certain boundedness to present daily precipitation. We concentrated on monthly runoff simulation and annual scale analysis, in order to reduce the uncertainty that brought by daily precipitation. The SWAT model achieved a good monthly runoff simulation on the large scale and long-term series, which is sufficient to support the study on the water balance component characteristics on the mean annual scale. This result can provide a credibility reference for basin water resource assessment and management. And most hydrological models are simulated monthly runoff in this study area, our research can be used to compare with the previous study. However, the monthly simulation hardly reflects the superiority of the gridded precipitation in spatial distribution. Thus, the further study on the water balance component characteristics on the inner-annual and small catchment scale is necessary. The hydrological model is widely used in the upper HRB to study hydrological processes, which NSEs are usually higher than 0.85 (Li et al. 2009, 2010, 2011; Lu et al. 2015). Compared with these studies using gauged precipitation, the simulation accuracy derived in the present study has yet to be improved. However, this study using high-resolution gridded precipitation, which is obviously superior to a few gauged station. The model calibration not only rely on hydrographs but also refer to basin features, such as base flow coefficient, evapotranspiration, snow melting runoff. Although the statistical evaluation criteria of simulation is not perfect, but the hydrological process and distribution of water balance components are more reasonable. The accuracy of the spatial distribution of water balance components has been improved. The 15 years simulation has a certain limitation to analysis water balance component changing trend. In this region, the meteorology and hydrology researches are plentiful and mature in historical period. Based on previous studies (Liu et al., 2012; Zang et al., 2013; Deng et al., 2015; Lu et al., 2015), we concentrated on the period of recent years, there few studies on this period. The underlving surface data used by SWAT model released in recent years, which is more credible for meteorology and hydrology changing trend analysis in recent years. For model climate forcing, only precipitation inputs used high-resolution gridded data; the temperature, wind speed, solar ration and relative humidity still used gauged data, which were scarce and unevenly distributed. The high-resolution gridded data of other climate elements should be applied in the SWAT model. For validation data, the precipitation lapse rate, soil water content and evapotranspiration lack gauged data match with the resolution of simulation, which is difficult to reflects the superiority of this study. These factors influence the accuracy of the model simulation. The runoff coefficient of the coniferous forest is relatively higher than those reported in previous studies; improvement of the calibration model is necessary (Gao et al., 2015; He et al., 2012; Yin et al., 2016). On the whole, the model parameters setting are empirical, the accuracy and resolution of validation data are too low and the study period is not long enough, which increases the uncertainty of model simulation. Future studies should focus on these limitations to investigate the SWAT model driven by high-resolution gridded data and to improve the model performance.

| Table 3. Water balance components for different regions. |                    |        |        |        |        |             |                  |
|----------------------------------------------------------|--------------------|--------|--------|--------|--------|-------------|------------------|
| Region                                                   | Area               | PREC   | ET     | WYLD   | SW     | Runoff      | Runoff           |
|                                                          | (km 2 ) | (mm/a) | (mm/a) | (mm/a) | (mm/a) | coefficient | contribution (%) |
| East tributary                                           | 2,504              | 609.8  | 364.4  | 229.9  | 63.5   | 0.37        | 29               |
| West tributary                                           | 5,032              | 522.8  | 310.3  | 199.5  | 58.5   | 0.38        | 52               |
| Main stream                                              | 2,482              | 446.2  | 287.4  | 148.3  | 27.7   | 0.33        | 19               |
| Entire basin                                             | 10,018             | 525.5  | 318.1  | 194.4  | 52.1   | 0.37        | 100              |

Fig. 1. Table 3. Water balance components for different regions

---

## Author Comment (AC2) · 20 Mar 2017

Thank you for your comments and suggestions. Those comments are all valuable and very helpful for revising and improving our paper, as well as important for further study of to our researches. We have studied comments carefully and have made correction which we hope meet with approval. The main corrections in the paper and the responds to comments are as flowing:

1) P1L12 "in inland river basins" is to general, add something like 'Tibet' Answer: Precipitation stations are usually scarce and unevenly distributed in inland river basins,

northeastern Tibetan Plateau.

2) P1L15 What kind of RCM is used, maybe better use "a" than "the", as there are many RCM out there. Answer: a regional climate model (RCM) is used to construct the soil and water assessment tool (SWAT).

3) P1L18 "upscale(d)"? Answer: The gridded precipitation is upscaled from the grid to the sub-basin scale and results in accurate representation of sub-basin precipitation input data.

4) P1L24 I expected at monthly scale larger NSEs Answer: Indeed, the NSEs are lower than similar hydrological simulation. But we calibrated model not only rely on hydrographs but also refer to basin features, such as base flow coefficient, evapotranspiration, snow melting runoff. Although the evaluation result is not perfect, but the simulation of hydrological process and water balance components are more reasonable.

5) P2L10 Again what kind of RCM?, better "a" than "the" Answer: RCM is the regional integrated environmental model system (RIEMS) developed by START TEA-COM RRC and Department of Atmospheric Science of Nanjing University. Revised in Section 1: The RIEMS regional climate model (RCM) can also supply information on spatial distribution to correct gridded precipitation (Xiong et al., 2013).

6) P2L12 "depict"? Answer: RCM simulation can well depict its spatial heterogeneity, which is suitable for driving the hydrological model (Wang et al., 2017).

7) P2L12 "A Soil and.", this sentence sounds like SWAT can only be applied in this basin Answer: RCM simulation can well depict its spatial heterogeneity, which is suitable for driving the hydrological model (Wang et al., 2017).

8) P2L45 "change trend" sounds strange to me, what is meant by a changing trend: the change of a change is no change? Answer: Finally, the gridded precipitation data and the hydrological simulation results were evaluated; and the spatial variability and

changing trend of water balance components were analysed.

9) P3L31 Your data is good? No Outliers? Did you check anything? Answer: All data is released by authorities, which is latest and open to the public. Many scholars have used these dataset to study meteorologic and hydrologic features in this region (Li et al., 2009, 2010, 2011; Yin et al., 2013, 2016; Lu et al., 2015, 2015; Qin et al., 2016; Wang et al., 2017). After we checking, found no problem. So the quality of the data is credible. 10) P4L8 At this point I asked myself about calibration and validation, you gave some answers to that later in chapter 4.4 Answer: In the section of method, we concentrate on the basic theory of the model. We present model setup in the section 4.4. But your suggestion is an desirable organization of paper structure.

11) P4L12 "Shepard interpolation" I had to google that, better known as "Inverse distance weighting" Answer: First, a gridded analysis of daily precipitation climatology was built based on the Inverse distance weighting interpolation method and mean daily gauged precipitation data from 1960 to 2014.

12) P4L13 where these station coming from, you said in chapter 2.2. that you only have 4 met. Station. Answer: The gridded precipitation data is produced by using meteorological stations and hydrological stations of the entire basin of the HRB. But this study concentrate on the upper HRB, so the gridded precipitation evaluation and model climate driven only involve 4 met. Station. In order to avoid misunderstanding, we have made correction as follows: The gauged precipitation, including meteorological stations and hydrological stations of the entire HRB, were smoothed by Fourier transformation to remove high-frequency noise precipitation caused by insufficient sampling, real extreme events and random measurement errors.

13) P4L15 "RCM was calibrated" How did you do this? Normally there are precipitation parametrisation schemes in RCMs, which more or less adequate model convective processes. Answer: The RIEMS RCM was develop by Xiong et al. (2013) and the gridded precipitation developed by Wang et al. (2017). These datasets were derived

СЗ

from HPSD. The gridded data directly employed RCM output. We did not produce gridded data, but we introduce the theory of gridded data in order to supplementary instruction of the data credibility. In order to avoid misunderstanding, we removed this sentence: Second, the RIEMS RCM provide the spatial distribution of the precipitation lapse rate (Xiong et al., 2013). The RCM simulation is used to correct the precipitation lapse rate of daily precipitation climatology.

14) P4L17 "optimal interpolation" who decided how what is optimal? Answer: The Optimal Interpolation (OI) is a interpolation method, which derived from Gandin. (1965). Revised as follow: Third, the Optimal Interpolation (OI) was employed to create the gridded ratio field that is the ratio of the daily gauged precipitation to daily precipitation climatology (Gandin., 1965).

15) P4L18 So at the end you did a residual correction, which means to me that at a pixel, which has a station in it, the observed value is matched by the simulated one? Am I right? Some kind of unbiasedness. Answer: The gridded data was produced by Wang et al (2017), we use this data as input for SWAT model. As you said, gridded daily precipitation is estimated as the product of the daily climatological precipitation and the daily precipitation ratio. The detailed introduction of interpolation method derived from by Wang et al (2017).

16) P5L21 "a" is the offset at 0m above ground, as precipitation will not be 0mm at this altitude Answer: "a" is the precipitation at the base location of subbasin.

17) P5L27 and L32 "Thus,.." and "Given the.." is unnecessarily doubling of sentences Answer: We have removed the sentence of "Thus...".

18) P5L28 Was this cross validation done by you? Did you really ensure that the two stations have not been used to interpolate the precipitation grids? Answer: The cross validation used meteorological station and hydrological station respectively. The detailed validation process can obtained from Wang et al (2017). Your concern is very significance. Considering it is not enough to validate the accuracy of entire gridded

data. Thus, we validated the spatial distribution of gridded data that close to the actual precipitation conditions in section 4.1.2.

19) P5L38 How did you compare the data? One pixel to one station? Or a mean of 3x3 pixels? Answer: Time series accuracy was evaluated by comparing gauged precipitation with the nearest pixel of gridded precipitation in a time series to assess the performance of the gridded data.

20) P6L2 "relatively high correlation" I cannot agree with that, there is no correlation at all. Your regression line is defined only by 0mm values, as they are obviously not excluded by this analysis. Tab1. The RMSE for daily precipitation is very large compared to the annual precipitation!! Answer: Indeed, using "relatively high correlation" is improper. Considering precipitation event has great uncertainty and randomness, and gridded data have a certain boundedness to present daily precipitation. So we removed the daily precipitation assessment. We concentrated on monthly runoff simulation and annual scale analysis, in order to reduce the uncertainty that brought by daily precipitation. Thus, we mainly assessment gridded precipitation data on monthly scale. We have discussed this question in Section 5. Revised as follow: Time series accuracy was evaluated by comparing gauged precipitation with the nearest pixel of gridded precipitation in a time series to assess the performance of the gridded data during the period of 2000 to 2014. Figs. 3 and 4 show the monthly comparison results in the Qilian and Yeniugou stations, respectively. The comparison of the monthly results indicated that the monthly gridded precipitation is close to the gauged precipitation, and their changing trends are consistent. Table 1 shows the criteria used to evaluate Qilian and Yeniugou. At the yearly scale, the mean annual gridded precipitation is close and slightly lower than the gauged data. The R2 values of Qilian and Yeniugou at the monthly scale all reached 0.99, indicating a strong correlation relationship. The daily monthly scale of PBIAS were controlled within  $\pm 1\%$  and the RMSE values were approximately 3 mm; the error was relatively low.

21) P6L8 These are not the real correlations, please exclude 0mm values. Answer:

We have removed the assessment of daily gridded precipitation.

22) P6L24 "was relatively overestimated" by your data? To which reference? Answer: Precipitation was slightly underestimated by the gridded data when comparing with gauged data.

23) P7L30 "most hydrological models (have)" Answer: most hydrological models have simulated monthly runoff in the upper HRB.

24) P7L31 "Thus, this..:" sounds not correct Answer: This study was simulated monthly runoff and evaluated on monthly scale, and the hydrological process was analysed from the monthly simulation results.

25) P7L36ff this sentence is not necessary, as I expected that from you Answer: We have removed this sentence.

26) P7L39 Why didn't you apply some auto calibration tool? Answer: There are many hydrological simulation and related research in this region, which provide reference for model parameter settling. The auto calibration may be achieve good simulations on hydrographs, but the hydrological process not necessarily close to actual situation. We are manually calibrated model not only rely on hydrographs but also refer to basin features, such as base flow coefficient, evapotranspiration, snow melting runoff. Although the assessment criteria of simulation is not perfect, but the hydrological process and distribution of water balance components are more reasonable. So we tend to manual calibration.

27) P8L13 "typical normal year: : :" that sounds strange to me Answer: The period when a river is at its normal level.

28) P8L26 "improve the accuracy.." I did not see any proof of that, you did not compare anything only one simulation and a observed time series. That is one of the main problems of your study I cannot judge your results to a former results as there is no comparison. Answer: This sentence is improper as you said, so we removed this sentence and discussed this question in Section 5. The hydrological model is widely used in the upper HRB to study hydrological processes, which NSEs are usually higher than 0.85 (Li et al. 2009, 2010, 2011; Lu et al. 2015). Compared with these studies using gauged precipitation, the simulation accuracy derived in the present study has yet to be improved. However, this study using high-resolution gridded precipitation which is obviously superior to a few gauged station. The model calibration not only rely on hydrographs but also refer to basin features, such as base flow coefficient, evapotranspiration, snow melting runoff. Although the statistical evaluation criteria of simulation is not perfect, but the hydrological process and distribution of water balance components are more reasonable. The accuracy of the spatial distribution of water balance components has been improved.

29) P8L33 "This finding: : :" that is the character of a balance, I did not expect something different, maybe you skip that. Answer: It indicates that the water balance components were relatively balanced.

30) P9L38 "mean annual change trends" what is meant by that. FIGURE 13 is unclear to me, what is shown there. How do you define a change trend? Answer: I'am so sorry to my careless, the figure was wrongly cited. The mean annual change trends means a long-term mean water balance at annual scale.

31) P10L13 "Thus,.." I could not follow you logic how is the simple precipitation connected to the drainage threshold? Answer: The relationship between drainage threshold and precipitation input data has been introduced in Section 4.2. Scale transformation by building virtual precipitation stations is important for grid upscale to sub-basin scale. The drainage area threshold of the sub-basin division is critical for scale transformation, which determines the distribution and amount of the sub-basin. Thus, the drainage area decide how many virtual precipitation station can be read into model. In other word, the drainage area decide the SWAT model how to make full use of highresolution gridded data.

32) P10L24 "superiority.." I couldn't see such a superiority of your daily data set! That was not shown in the manuscript sorry! Answer: We emphasis on the superiority of monthly scale and spatial distribution: However, the monthly simulation hardly reflects the superiority of the gridded precipitation in spatial distribution. Thus, the further study on the water balance component characteristics on the inner-annual and small catchment scale is necessary. Revised in Section 5: The hydrological model is widely used in the upper HRB to study hydrological processes, which NSEs are usually higher than 0.85 (Li et al. 2009, 2010, 2011; Lu et al. 2015). Compared with these studies using gauged precipitation, the simulation accuracy derived in the present study has yet to be improved. However, this study using high-resolution gridded precipitation, which is obviously superior to a few gauged station. The model calibration not only rely on hydrographs but also refer to basin features, such as base flow coefficient, evapotranspiration, snow melting runoff. Although the statistical evaluation criteria of simulation is not perfect, but the hydrological process and distribution of water balance components are more reasonable. The accuracy of the spatial distribution of water balance components has been improved.

33) P10L29 "The temperature: : :" The RCM should deliver these variables to, they are even more reliable than precipitation from RCMs, so why didn't you use them? Answer: In this study, we concentrate on the study of high-resolution precipitation to improve runoff simulation. But the gridded data and RCM data are belong to different category. If using two kinds of data at the same time will add the uncertainty. It is hamper to discussed the precipitation data how to influence model simulation. Precipitation and temperature are the primary driver of the hydrological processes in a basin. Your advice is perfect, we will add the same kind of temperature in the future work. Thus, we discussed this question in Section 5: For model climate forcing, only precipitation inputs used high-resolution gridded data; the temperature, wind speed, solar ration and relative humidity still used gauged data, which were scarce and unevenly distributed. The high-resolution gridded data of other climate elements should be applied in the SWAT model.

34) P11L3 and L9 " exhibits high time series accuracy" and "superiority" that was not shown! Answer: We emphasis on the superiority of gridded precipitation on monthly scale and spatial distribution. Revised as follows: The time series accuracy of monthly gridded precipitation were assessed in Qilian and Yeniugou. The RMSE is approximately 3 mm and the PBIAS is controlled within  $\pm 1\%$ ; both exhibit a strong correlation. The spatial distribution of gridded precipitation decreased from the southeast to the northwest and increased with elevation, which consisted of real precipitation features. Thus, these datasets exhibited high time series accuracy on monthly scale and spatial description ability in the study area.

35) P11L25 "trends" In terms of climatology the 14 years you investigated are too short for a trend analysis. You need at least >30a as your observed trends maybe only caused by natural climate variability. Answer: In this region, the meteorology and hydrology researches are amounts and mature in historical period. So we are fairly comprehend climatology in historical period. Based on previous studies, we concentrate on meteorology and hydrology study in recent years, there are few scholar conduct studies in this period. The underlying surface data used by this study released in recent years, which is more credible for meteorology and hydrology changing trend analysis in recent years. But your advice is very constructive. In the future work, we will use multi-period underlying surface data. Analyzing long-term changing trend of meteorology and hydrology features. Revised in Section 5: The 15 years simulation has a certain limitation to analysis water balance component changing trend. In this region, the meteorology and hydrology researches are plentiful and mature in historical period. Based on previous studies, we concentrate on the period of recent years, there are few studies on this period. The underlying surface data used by SWAT model released in recent years, which is more credible for meteorology and hydrology changing trend analysis in recent years.

| Table 1. Calculation of the evaluation criteria. |                      |                        |               |           |           |
|--------------------------------------------------|----------------------|------------------------|---------------|-----------|-----------|
| Station                                          | Yearly scale         |                        | Monthly scale |           |           |
|                                                  | Gauge data
(mm/a) | Gridded data
(mm/a) | $R^2$         | RMSE (mm) | PBIAS (%) |
| Qilian                                           | 424.5                | 422.8                  | 0.99          | 2.73      | 0.62      |
| Yeniugou                                         | 460.4                | 459.0                  | 0.99          | 2.57      | 0.30      |

**Fig. 2.** Figure 3. Scatter diagram of monthly gauge precipitation and monthly gridded precipitation in Qilian (a) and Yeniugou (b) (The red line is trend line)

Fig. 3. Figure 14. Changing trend of the water balance components at the sub-basin scale.